# Age-Related Changes in Episodic Processing of Scenes: A Functional Activation and Connectivity Study

**DOI:** 10.3390/s23084107

**Published:** 2023-04-19

**Authors:** Makoto Miyakoshi, Josephine Astrid Archer, Chiao-Yi Wu, Toshiharu Nakai, Shen-Hsing Annabel Chen

**Affiliations:** 1Division of Child and Adolescent Psychiatry, Cincinnati Children’s Hospital Medical Center, Cincinnati, OH 45229, USA; makoto.miyakoshi@cchmc.org; 2Department of Psychiatry, University of Cincinnati College of Medicine, Cincinnati, OH 45267, USA; 3Department of Gerontechnology, National Center for Geriatrics and Gerontology, Ohbu 474-8511, Aichi, Japan; 4School of Social Sciences, Nanyang Technological University, Singapore 639818, Singapore; jomail@gmx.co.uk; 5Centre for Research in Child Development, National Institute of Education, Nanyang Technological University, Singapore 637616, Singapore; chiao-yi.wu@nie.edu.sg; 6Department of Dental Radiology, Graduate School of Dentistry, Osaka University, Suita 565-0871, Osaka, Japan; nakai.niinf@dent.osaka-u.ac.jp; 7Institute of NeuroImaging & Informatics, Ohbu 474-8511, Aichi, Japan; 8Centre for Research and Development in Learning, Nanyang Technological University, Singapore 637335, Singapore; 9Lee Kong Chian School of Medicine, Nanyang Technological University, Singapore 308232, Singapore

**Keywords:** posterior-to-anterior-shift in aging, compensatory, episodic processing, scenes, inferior frontal gyrus, hippocampus, parahippocampus

## Abstract

The posterior-to-anterior shift in aging (PASA) effect is seen as a compensatory model that enables older adults to meet increased cognitive demands to perform comparably as their young counterparts. However, empirical support for the PASA effect investigating age-related changes in the inferior frontal gyrus (IFG), hippocampus, and parahippocampus has yet to be established. 33 older adults and 48 young adults were administered tasks sensitive to novelty and relational processing of indoor/outdoor scenes in a 3-Tesla MRI scanner. Functional activation and connectivity analyses were applied to examine the age-related changes on the IFG, hippocampus, and parahippocampus among low/high-performing older adults and young adults. Significant parahippocampal activation was generally found in both older (high-performing) and young adults for novelty and relational processing of scenes. Younger adults had significantly greater IFG and parahippocampal activation than older adults, and greater parahippocampal activation compared to low-performing older adults for relational processing—providing partial support for the PASA model. Observations of significant functional connectivity within the medial temporal lobe and greater negative left IFG-right hippocampus/parahippocampus functional connectivity for young compared to low-performing older adults for relational processing also supports the PASA effect partially.

## 1. Introduction

Aging is generally associated with poorer memory encoding [1] and retrieval [2]. However, there are individual differences in this aging effect, and high-functioning older adults can show comparable memory performance as their young counterparts [3,4]. Clarifying the mechanism of the aging effect on memory functions and the sources of the individual differences is important to find an effective intervention to maintain QOL of the older people.

Findings from human neuroimaging studies using positron emission tomography (PET) and functional MRI (fMRI) showed that when older adults performed episodic encoding and retrieval tasks, their brain activation shows a general pattern of activation decrease in the posterior part of the brain combined with activation increase in the anterior part of the brain [5,6]. Based on this observation, the posterior-to-anterior shift in aging (PASA) model was proposed [7]. Results from a functional connectivity study also supported this view [8]. The widely accepted view of the functional role of the PASA effect today is active compensation [9]. There is empirical evidence that the posterior to anterior shift was associated with increased cognitive demands [10] and the PASA effect benefitted older adults in episodic encoding [6].

The conventional PASA model has been focusing on frontal and occipital brain regions. Here, we claim that the PASA model may be extended to the relation between the hippocampal formation and the frontal regions, particularly the inferior frontal gyrus (IFG). We found it suggestive of the extended PASA model that an age-related increase in bilateral IFG activation coupled with a decrease in bilateral hippocampal and parahippocampal activation for novelty processing of the scenes (i.e., experimentally defined as watching novel pictures minus repeating pictures; see [11]) and relational processing of the scenes (watching novel pictures minus scrambled pictures) [12,13,14]. Being motivated by these findings, we designed a study to test the central hypothesis of whether such hippocampal-frontal (i.e., temporal-to-frontal) shift in aging should offer any compensatory effect for the older adults during novelty and relational processing of scenes. We employed an established paradigm developed to examine the novelty and relational processing using indoor/outdoor scenes [11,15], a part of which was published elsewhere [15]. In conducting the analysis, we controlled age-related structural atrophy in the prefrontal cortex (PFC) [16], hippocampus [14], and parahippocampus [17]. We evaluated whether age-related changes in the BOLD signals reflect the PASA effect within bilateral IFG, hippocampi, and parahippocampi as regions of interest (ROIs) in comparison between healthy young and older adults. The older adults were further separated into two subgroups, high and low performers, to study the compensatory role of the PASA model.

## 2. Method

### 2.1. Participants

A total of 102 older and young adults were separately recruited in this study by the Clinical Brain Lab (CBL) in Singapore and National Center for Geriatrics and Gerontology (NCGG) in Japan. Twenty-five older adults (12 females, 3 left-handed, mean age = 67.8, *SD* = 7.6) with a mean mini-mental state exam (MMSE) [18] score of 28.9 (*SD* = 1.1, range 26–30) and 25 young adults (12 females, 2 left-handed, mean age = 23.2, *SD* = 2.2) with a mean MMSE score of 29.9 (*SD* = 0.3, range 29–30) were recruited by CBL. Twenty-six older adults (13 females, 2 left-handed, mean age = 66.7, *SD* = 3.1) with a mean mini mental status examination (MMSE) score of 28.3 (*SD* = 1.9, range 23–30) and 26 young adults (14 females, 1 left-handed and 1 ambidextrous, mean age = 21.7, *SD* = 1.6) with a mean MMSE score of 29.2 (*SD* = 0.9, range 27–30) were recruited by NCGG. The Edinburgh Handedness Inventory was used to determine participants’ handedness [19]. All participants were screened to exclude any psychiatric, neurologic, and metabolic conditions (e.g., diabetes and hypertension). Participants with excess head motion (>2 mm translation, >3 degrees rotation), poor response accuracy (ACC) (<60% ACC during scan and post-scan), and low MMSE score (MMSE score ≤ 26) were excluded from the analyses. Both CBL and NCGG datasets were combined and analyzed for age-related changes in functional activation, while only the NCGG dataset was analyzed for age-related changes in functional connectivity for novelty and relational processing of scenes.

The older adults were further categorized into low and high performers based on their memory performance ACC for novelty and relational processing of scenes. Unlike Cabeza et al. [3], we did not administer standardized cognitive tests other than MMSE. Hence, we categorized older adults into older low and older high performers using the mean ACC of young adults on our task, with a 2 standard deviation (*SD*) buffer, as the cut-off in the respective NCGG and CBL datasets. Thus, older adults were considered as older high performers if their ACC fell within 2 *SD* from the mean ACC of their young counterparts, while the older low performers had ACC beyond 2 *SD* from the mean ACC of their young counterparts in the respective datasets. Older high performers had comparable ACC as young performers, while older low performers showed significantly poorer ACC than older high and young performers.

In total, 33 older adults (19 females, 3 left-handed, mean age = 66.2, *SD* = 5.0) with a mean MMSE score of 29.0 (*SD* = 0.9, range 27–30), and 48 young adults (26 females, 2 left-handed and 1 ambidextrous, mean age = 22.4, *SD* = 2.1) with a mean MMSE score of 29.5 (*SD* = 0.8, range 27–30), were analyzed for age-related changes in functional activation related to novelty and relational processing of scenes. Among the older adults, there were 9 older low performers (7 females, 2 left-handed, mean age = 69.6, *SD* = 5.8) with a mean MMSE score of 28.7 (*SD* = 1.1, range 27–30) and 24 older high performers (12 females, 1 left-handed, mean age = 64.9, *SD* = 4.1) with a mean MMSE score of 29.1 (*SD* = 0.8, range 28–30). Although the older low performers were significantly older than older high performers, *t*(31) = 2.58, *p* < 0.05; the MMSE score for older low performers and older high low performers did not differ significantly, *t*(31) = 1.32, *p* > 0.05.

For age-related changes in functional connectivity related to novelty and relational processing of scenes, a total of 17 older adults (10 females, 1 left-handed, mean age = 66.1, *SD* = 3.4) with a mean MMSE score of 28.8 (*SD* = 1.0, range 27–30), and 25 young adults (14 females, 1 ambidextrous, mean age = 21.7, *SD* = 1.7) with a mean MMSE score of 29.2 (*SD* = 0.9, range 27–30), from the NCGG dataset were analyzed. Among the older adults, there were 6 older low performers (4 females, 1 left-handed, mean age = 67.2, *SD* = 2.5) with a mean MMSE score of 28.7 (*SD* = 1.4, range 27–30) and 11 older high performers (6 females, 0 left-handed, mean age = 65.5, *SD* = 3.8) with a mean MMSE score of 28.8 (*SD* = 0.9, range 28–30). These older low performers did not differ significantly in an age when compared to older high performers, *t*(15) = 0.92, *p* > 0.05; and the MMSE score for older low performers and older high low performers did not differ significantly, *t*(15) = 0.28, *p* > 0.05.

Written informed consent, which had been approved by both the ethics committees of NCGG, Japan, and Nanyang Technological University, Singapore, in accordance with the Helsinki Declaration, was obtained from all participants.

### 2.2. Task

The investigation of episodic encoding in terms of stimulus novelty and relational processing was adapted from Binder et al.’s protocols [11]. These used an indoor/outdoor scene discrimination task, sensitive to novelty and relational processing. There were three conditions in this indoor/outdoor scene discrimination task, namely Novel Pictures condition (N), Repeating Pictures condition (R), and Scrambled Pictures condition (S). For both Novel Pictures and Repeating Pictures conditions, participants were asked to discriminate whether each non-scrambled picture was an indoor or outdoor scene. In the Scrambled Pictures condition, participants were asked to discriminate whether the left and right halves of each scrambled picture were identical.

Colored digital indoor/outdoor scenes (containing no words or people), re-sized to 600 × 600 pixels, subtended 22.5° of horizontal and vertical visual angle with a scanner, were used for stimulus presentation in the Novel Pictures and Repeating Pictures conditions. For the Novel Pictures condition, a total of 40 unique pictures (20 indoor and 20 outdoor scenes) were presented to each participant who had to discriminate whether the picture was an indoor scene with a button by the index finger or an outdoor scene with another button by the middle finger. In the Repeating Pictures condition, participants were presented with two unique pictures (one indoor and one outdoor scene) repeatedly for a total of 20 times. They pressed a button assigned to either index or middle finger as above to discriminate the indoor/outdoor scenes.

For the Scrambled Pictures condition, pictures of indoor/outdoor scenes were divided into halves, defined by a red line that bisected each picture into two equal hemifields. Pictures of indoor/outdoor scenes were scrambled into pixelated mosaics by re-arranging 20-pixel square segments. The 600 × 300 pixelated halves of these scrambled indoor/outdoor scenes were either duplicated to produce scrambled pictures with identical halves or combined with another retiling of the same picture of the indoor/outdoor scene to produce scrambled pictures with non-identical halves. A total of 40 unique scrambled pictures (20 scrambled pictures with identical halves and 20 scrambled pictures with non-identical halves) were presented to each participant who had to discriminate whether both halves of the scrambled picture were identical with a button press by the index finger, or non-identical with the middle finger.

The stimuli presentation for the indoor/outdoor scenes discrimination task was blocked by condition, in the order of Novel Pictures, followed by Repeating Pictures, then Scrambled Pictures condition. Each block consisted of eight trials. For each trial, a stimulus was presented against a black background for 2500 ms, followed by 500 ms of inter-stimulus interval. Participants had to complete a total of five cycles per run which lasted for 240 s (see Figure 1). Practice trials were given before the scan to familiarize participants with the task. Behavioral performance in terms of accuracy rate (ACC) and reaction time (RT) for the indoor/outdoor scene discrimination task were measured for all three conditions during the scan.

A post-scan recognition test measuring participants’ ACC on non-scrambled and scrambled scenes was administered (while participants were still in the scanner) to participants in the CBL dataset. This post-scan recognition test consisted of Non-scrambled Pictures and Scrambled Pictures conditions. For the Non-scrambled Pictures condition, 15 old (presented previously during the scan) and 15 new (not presented previously during the scan) non-scrambled scenes were presented to each participant who had to recognize whether they had seen the non-scrambled pictures before during the scan, via button pressing. For the Scrambled Pictures condition, 15 old (presented previously during the scan) and 15 new (not presented previously during the scan) scrambled scenes were presented to each participant who had to recognize whether they had seen the scrambled pictures before during the scan, via button pressing. This post-scan recognition test would serve as a manipulation check that stimulus novelty (novelty encoding) and relational processing (relational encoding) of scenes were previously elicited by the indoor/outdoor scenes discrimination task.

As outlined by Binder et al. [11], a subtraction analysis of the Novel Pictures minus Repeating Pictures (N > R) contrast would resemble novelty processing of scenes, while the Novel Pictures minus Scrambled Pictures (N > S) contrast would emphasize relational processing of scenes. Both novelty and relational processing were incidental in nature as the participants were not explicitly instructed to remember any stimuli; neither were participants in the CBL dataset informed of a post-scan recognition test.

### 2.3. Image Acquisition and Preprocesses

Structural and functional scans were acquired using a 3.0-Tesla MRI scanner (MAGNETOM Trio, Siemens, Erlangen, Germany) in both centers, as participants underwent the indoor/outdoor scenes discrimination task. VisuaStimDigital for MRI (Resonance Technology Inc., CA, United States) was used for the presentation of stimuli.

Functional imaging acquisition followed a gradient-echo echo-planar sequence for both datasets: CBL (TE, 30 ms; TR, 3000 ms; FOV, 192 mm; matrix, 64 × 64; slice thickness, 3 mm; axial slices, 39; gap between slices 0.75 mm); NCGG (TE, 24 ms; TR, 2000 ms; FOV, 192 mm; matrix, 64 × 64; slice thickness, 3 mm; axial slices, 39; gap between slices 0.75 mm). A total of 120 and 180 consecutive image volumes were acquired per scan per participant for the CBL and NCGG datasets, respectively. Participants’ T1-weighted anatomical reference image was used for localization purposes functional images were acquired at different sites, and inter-scanner variability was controlled by submitting the signal-to-fluctuation-noise-ratios (SFNRs) [20] extracted from each participant as covariates in the group-level imaging analyses [21].

Preprocessing and functional activation analyses were performed using statistical parametric mapping [22] in SPM8 (The FIL Methods Group, Wellcome Trust Centre for Neuroimaging, Institute of Neurology, UCL), with MATLAB 7.9 (The MathWorks, Natick, MA, USA). For each participant, the T1-weighted anatomical image and functional images were re-oriented to have the anterior and posterior commissures on the same plane, using the T1-weighted image as the image template.

Preprocessing of each participant’s functional images followed the DARTEL (i.e., Diffeomorphic Anatomical Registration using Exponentiated Lie algebra) pipeline [23], consisting of realignment, slice-timing correction, and structural-to-functional co-registration. DARTEL was used to improve inter-subject structural-to-functional co-registration [24]. Using the group template and the individual participant’s flow field image created from DARTEL, each participant’s T1-weighted anatomical image, functional images, and gray matter probability map were normalized to Montreal Neurological Institute space (MNI; Montréal, QC, Canada).

The gray matter probability maps generated from DARTEL enabled us to control for any age-related structural atrophy [25], by submitting them as covariates in the functional activation analyses. For functional connectivity analyses, intra-cranial brain volumes were submitted as covariates to control for age-related structural differences.

Functional images were later smoothed with a Gaussian kernel of 8 mm full width at half maximum (FWHM). To prevent attenuation of SFNR caused by smoothing, a separate copy of unsmoothed functional images for each participant was saved for SFNR extraction.

### 2.4. Design for Statistical Tests

#### 2.4.1. Behavioral Data

For behavioral analyses, a 2 (Age: Older, Young) × 3 (Condition: Novel Pictures, Repeating Pictures, Scrambled Pictures) mixed design analysis of variance (ANOVA) was conducted on participants’ ACC and RT. In addition, a 3 (Performance: Older Low Performers, Older High Performers, Young Performers) × 3 (Condition: Novel Pictures, Repeating Pictures, Scrambled Pictures) ANOVA was conducted on participants’ ACC and RT. For the CBL dataset, a 2 (Age: Older, Young) × 2 (Condition: Non-scrambled Pictures, Scrambled Pictures) mixed design ANOVA was conducted on post-scan ACC. This served as a manipulation check that the incidental encoding in terms of novelty and relational processing were indeed elicited. The Greenhouse–Geisser correction was used whenever necessary.

Post-hoc multiple pair-wise comparisons, with Bonferroni correction, were carried out when a significant interaction or main effect was found, with *p* < 0.05. As a significant age effect on ACC was found, participants’ ACCs from the respective conditions were entered as covariates in the group-level imaging analyses.

#### 2.4.2. Imaging Data

Functional activation analyses. For functional activation analyses, general linear model analyses were separately conducted for N > R and N > S contrasts using SPM8, to generate subject-level brain activation maps for novelty and relational processing of scenes, respectively. The subject-level brain activation maps were subsequently submitted for group-level random effects analyses, while controlling for gray matter probability, SFNRs, and ACCs, in biological parametric mapping (BPM) [26]. The significant level for cluster-level activation was set at *p* < 0.05 (*FDR*-corrected), with cluster size *k* ≥ 20 voxels.

Functional connectivity analyses. ROI-to-ROI functional connectivity analyses were conducted using the functional connectivity toolbox (CONN) [27,28]. As the scanning parameters used for NCGG (TR = 2000 ms) and CBL (TR = 3000 ms) datasets were different, we could not merge both datasets in CONN. Thus, we selected the NCGG dataset to be used in CONN for functional connectivity analyses, as the NCGG dataset allowed a less disproportional splitting of the older adults into low and high performers compared to the CBL dataset. In the Setup phase, individual subjects’ intracranial brain volume and ACCs were submitted as second-level covariates for ROI-to-ROI functional connectivity analysis, with left and right IFG, hippocampus, and parahippocampus as a priori ROIs. Next, in the Denoising phase, a bandpass filter of 0.0085 to 0.16 Hz was applied. Subject-level brain activation results were then generated in the First-level Analyses phase with left and right IFG, hippocampus, and parahippocampus as a priori ROIs. Subsequently, the subject-level brain activation results were submitted for group-level ROI-to-ROI functional connectivity analyses, while controlling for intracranial brain volume and ACCs. Group-level ROI-to-ROI functional connectivity focused on left and right IFG, hippocampus, and parahippocampus were generated, with N > R and N > S contrasts for novelty and relational processing of scenes, respectively, at *p* < 0.05 (*FDR*-corrected).

Regions of interest. Left and right IFG, hippocampus, and parahippocampus masks were created using MARINA software [29], and the left–right inversion of the masks were later corrected using the SPM toolbox, MarsBaR [30]. These left and right IFG, hippocampus, and parahippocampus masks would serve as ROIs in the ROI-to-ROI functional connectivity analyses for novelty and relational processing of scenes.

## 3. Results

### 3.1. Behavioral Data

A 2 (Age: Older, Young) × 3 (Condition: Novel Pictures, Repeating Pictures, Scrambled Pictures) ANOVA on ACC revealed significant age × condition interaction effect, *F*(2, 79) = 8.04, *p* < 0.05, *η*^2^ = 0.09, and a significant main effect of condition, *F*(2, 79) = 47.66, *p* < 0.05, *η*^2^ = 0.38. Post-hoc analyses showed that the older adults on average performed significantly poorer than young adults for the Scrambled Pictures condition (90.69% vs. 94.50%), *p* < 0.05. On average, the Scrambled Pictures condition had significantly poorer performance than Novel Pictures or Repeating Pictures condition for both older (90.39% vs. 97.09%, 90.39% vs. 98.09%, respectively) and young adults (94.50% vs. 97.02%, 94.50% vs. 97.92%, respectively), *p* < 0.05 (see Figure 2a). Other pairwise comparisons did not reveal significant differences in ACC.

A similar 2 × 3 ANOVA on RT showed significant age × condition interaction effect, *F*(2, 79) = 14.50, *p* < 0.05, *η*^2^ = 0.89, and significant main effects of age and condition, *F*(1, 79) = 19.87, *p* < 0.05, *η*^2^ = 0.20 and *F*(2, 79) = 644.76, *p* < 0.05, *η*^2^ = 0.89, respectively. Post-hoc analyses showed that older adults on average performed significantly slower than young adults for the Novel Pictures condition (894.35 ms vs. 778.60 ms), Repeating Pictures condition (701.82 ms vs. 651.67 ms), and Scrambled Pictures condition (1360.81 ms vs. 1135.64 ms), *p* < 0.05. The Scrambled Pictures condition, on average, had the slowest performance followed by the Novel Pictures condition then the Repeating Pictures condition, for both older and young adults, *p* < 0.05 (see Figure 2b).

Upon separating the older adults into high and low performers, a 3 (Performance: Older Low Performers, Older High Performers, Young Performers) × 3 (Condition: Novel Pictures, Repeating Pictures, Scrambled Pictures) ANOVA on ACC revealed significant performance × condition interaction effect, *F*(3, 79) = 9.92, *p* < 0.05, *η*^2^ = 0.20, and significant main effects of performance and condition, *F*(2, 79) = 17.18, *p* < 0.05, *η*^2^ = 0.31 and *F*(2, 79) = 65.29, *p* < 0.05, *η*^2^ = 0.46, respectively. Post-hoc analyses showed that older low performers on average performed significantly poorer than both older high performers and young performers for the Repeating Pictures condition (94.44% vs. 99.38%, 94.44% vs. 97.81%, respectively) and Scrambled Pictures condition (81.05% vs. 93.72%, 81.05% vs. 94.32%, respectively), *p* < 0.05 (see Figure 3a). Both older high performers and young performers did not differ significantly in their performance for all three conditions, *p* > 0.05.

A similar 3 × 3 ANOVA on RT demonstrated a significant performance × condition interaction effect, *F*(3, 79) = 7.18, *p* < 0.05, *η*^2^ = 0.16, and significant main effects of performance and condition, *F*(2, 79) = 10.62, *p* < 0.001, *η*^2^ = 0.21 and *F*(2, 79) = 465.01, *p* < 0.05, *η*^2^ = 0.86, respectively. Post-hoc analyses showed that both older low performers and older high performers on average performed significantly slower than young adults for the Novel Pictures condition (942.31 ms vs. 778.60 ms, 876.37 ms vs. 778.60 ms, respectively) and Scrambled Pictures condition (1400.59 ms vs. 1135.64 ms, 1345.89 ms vs. 1135.64 ms, respectively), *p* < 0.05 (see Figure 3b). No significant difference in RT was found in other pair-wise comparisons across the performance.

A 2 (Age: Older, Young) × 2 (Condition: Non-scrambled Pictures, Scrambled Pictures) mixed design ANOVA, conducted on post-scan ACC for the CBL dataset, revealed significant age × condition interaction effect, *F*(1, 37) = 4.47, *p* < 0.05, *η*^2^ = 0.11, and significant main effects of age and condition, *F*(1, 37) = 6.11, *p* < 0.05, *η*^2^ = 0.14 and *F*(1, 37) = 140.69, *p* < 0.05, *η*^2^ = 0.79, respectively. Post-hoc analyses showed that the older adults on average performed poorer than young adults on post-scan ACC for the Non-scrambled Pictures condition (72.38% vs. 80.65%), *p* < 0.05, but not for the Scrambled Pictures condition, *p* > 0.05. On average, the Scrambled Pictures condition had poorer performance than the Non-scrambled Pictures condition for both older (72.38% vs. 56.00%) and young adults (80.65% vs. 57.17%), *p* < 0.05 (see Figure 4). It is clear that the Age × ACC interaction is dominated by the simple effect of Non-scrambled Pictures rather than the Scrambled Pictures between the Older and Young adults, which indicate the task’s sensitivity to separate the two Age groups according to the scene encoding but not to geometric pattern encoding.

As a significant age effect of ACC was found, the ACCs measured for all conditions were submitted as non-imaging covariates during group-level imaging analyses for novelty processing of scenes.

### 3.2. Functional Activation Analyses

Novelty processing of scenes. Results of supra-threshold cluster-level activations, at *p* < 0.05 (*FDR*-corrected) with *k* ≥ 20 for N > R contrast representing novelty processing of scenes, are shown in Table 1. Participants’ gray matter probability, SFNR value, and ACC in both Novel Pictures and Repeating Pictures conditions were controlled as covariates in group-level analyses. Significant cluster-level activations related to novelty processing of scenes for older adults (*n* = 33) were observed in the right superior frontal, precentral and postcentral gyri, left superior temporal, middle temporal, and fusiform gyri, bilateral parahippocampal gyrus, and right cerebellar tonsil (see Figure 5a). Among the older adults, only the older high performers (*n* = 24) showed significant activations in the right middle frontal, precentral, postcentral, and superior temporal gyri, left middle temporal and fusiform gyri, right hippocampus, left parahippocampal gyrus, and right cerebellum Lobule 8/9 (see Figure 5b). No significant activations were found for older low performers. In contrast, significant cluster-level activations related to novelty processing of scenes for young adults (*n* = 48) were observed in the right middle frontal and fusiform gyri, right hippocampus, and left parahippocampal gyrus (see Figure 5c). Group-level comparisons among the different age or performance groups for novelty processing of scenes did not reveal significant differences in functional activations.

Relational processing of scenes. Results of supra-threshold cluster-level activations, at *p* < 0.05 (*FDR*-corrected) with *k* ≥ 20 for N > S contrast representing novelty processing of scenes, are shown in Table 2. Participants’ gray matter probability, SFNR value, and ACC in both Novel Pictures and Scrambled Pictures conditions were controlled as covariates in group-level analyses. Significant cluster-level activations related to novelty processing of scenes for older adults (*n* = 33) were observed in the left superior frontal gyrus, bilateral superior temporal gyrus, left middle temporal and angular gyri, right middle temporal gyrus, bilateral insula, left cingulate gyrus and anterior cingulate, bilateral posterior cingulate, left hippocampus, and bilateral parahippocampus gyrus (see Figure 6a). Among the older adults, only the older high performers (*n* = 24) showed significant activations in the left superior frontal gyrus, bilateral superior temporal gyrus, left middle temporal gyrus, right inferior temporal gyrus, left angular and superior occipital gyri, bilateral insula, left cingulate gyrus, bilateral anterior and posterior cingulate, and bilateral parahippocampal gyri (see Figure 6b). No significant activations were found for older low performers. In contrast, significant cluster-level activations related to relational processing of scenes for young adults (*n* = 48) were observed in the right inferior frontal gyrus, left middle temporal gyrus, posterior cingulate and parahippocampal gyrus, bilateral cerebellum Crus II, right cerebellum Crus I, and right cerebellum Lobule 9 (see Figure 6c). BA, Brodmann Area.

Age-related differences in functional activation have also been found for the relational processing of scenes. Young adults versus older adults (*n* = 81) revealed significantly greater activations in the bilateral superior frontal and middle frontal gyri, left inferior frontal gyrus, bilateral medial frontal gyrus, left precentral gyrus, right inferior parietal lobule, bilateral precuneus, right superior temporal gyrus, bilateral middle temporal gyrus, cuneus, and lingual gyrus, left cingulate gyrus, right anterior cingulate, left posterior cingulate, bilateral parahippocampal gyrus, left fusiform gyrus, right cerebellum Crus I, right cerebellum Lobule 7b, left cerebellum Crus II, right cerebellum Lobule CrusI/8/9 (see Figure 6d). In particular, when compared against older low performers (*n* = 57), young performers showed significantly greater activations in the bilateral parahippocampal gyrus (see Figure 6e). BA, Brodmann Area.

### 3.3. Functional Connectivity Analyses

Novelty processing of scenes. ROI-to-ROI functional connectivity analyses were conducted with a priori ROIs, namely left and right IFG, hippocampus, and parahippocampus, at *p* < 0.05 (*FDR*-corrected), for N > R contrast representing novelty processing of scenes. Participants’ intracranial brain volume and ACC in both Novel Pictures and Repeating Pictures conditions were controlled as covariates in group-level analyses. However, no significant functional connectivity was found among these ROIs.

Relational processing of scenes. ROI-to-ROI functional connectivity analyses were conducted with a priori ROIs, namely left and right IFG, hippocampus, and parahippocampus at *p* < 0.05 (*FDR*-corrected) for N > S contrast representing relational processing of scenes. Results are shown in Table 3. Participants’ intracranial brain volume and ACC in both Novel Pictures and Scrambled Pictures conditions were controlled as covariates in group-level analyses. Significant positive left parahippocampus–right parahippocampus functional connectivity was found for older low performers (*n* = 6) (see Figure 7a). In contrast, young performers (*n* = 25) revealed significant positive left hippocampus–right hippocampus functional connectivity (see Figure 7b). In terms of group comparisons, marginal significance was found between young and older low performers for the relational processing of scenes. In particular, young performers demonstrated greater negative left IFG–right hippocampus and left IFG–right parahippocampus functional connectivity than older low performers (see Figure 7c). It is noteworthy that the age-related increase, not decrease, of the functional connectivity was found in the low-performing older adults, a pattern predicted by the PASA model.

## 4. Discussion

We predicted that the PASA model may be extended to the relation between the hippocampal formation and the frontal regions, particularly the inferior frontal gyrus (IFG). Our prediction was partially supported by the results from the present study: In the relational processing, age-related changes in activations and functional connectivity between IFG (i.e., top-down modulation) and hippocampal/parahippocampal structures (i.e., bottom-up processing) were identified to be performance-related for the old adults (see Young > Old-Low adults in Table 2 and Table 3). The detail will be discussed below.

### 4.1. The Extended PASA Model between IFG and Hippocampus/Parahippocampus

During the relational task, the low-performing older adults showed less parahippocampal activations but more functional connectivity between the hippocampal/parahippocampal and IFG compared with the young adults. These differences were not found when comparing the high-performing older adults with the young adults. We conclude that the parahippocampal modulation found in the low-performing older adults is in line with the posterior aspect of the PASA model, namely the decrease in activations in the aging brain. We also conjecture this parahippocampal alteration in low-performing older adults may be interpreted in relation to compensation because positive prefrontal-hippocampus functional connectivity is usually associated with better memory performance [31]. However, it does not seem to provide convincing support for the compensation view. We will revisit this problem later.

Some of the past literature has shown activations in the IFG, hippocampus, and parahippocampus were found for both older and young adults during novelty and relational processing of scenes [11,13,14,32]. Our results are consistent with these findings. Furthermore, the intact parahippocampal activation in high-performing older adults and the lack of it in low-performing older adults in our results validated the view that the parahippocampus subserves the encoding process [13,33] as well as the associative processing [34] of scenes.

### 4.2. The Frontal Aspect of the PASA Model Was Not Replicated

In the current study, we did not observe the activation increase in the IFG in the older adults compared with the young adults, although this aging-related increase in the anterior brain is predicted as a part of the PASA effect. Even after separating the older adults into high- and low-performance subgroups, the predicted age-related increase was not found. Thus, our results did not replicate the previous studies [12,13,14,35].

We observed that young but not older adults showed significant activation in the IFG during the relational processing task. The role of IFG, an area in the PFC, during this task, is to maintain the encoded information in a top-down manner [36]. The IFG activation observed in our study may suggest that the young adults could maintain the encoded information better than the older adults. This is consistent with the fact that functional connectivity between PFC and MTL is required for successful memory encoding [36,37,38,39]. One possible explanation for not observing the expected activation increase in IFG in the older adults is that the task design for the encoding process did not work as intended. The detail of this interpretation will be discussed in the Limitation section. Another possibility is that the anterior aspect of the PASA model is less robust than the posterior aspect of the PASA model.

### 4.3. Compensation vs. Dedifferentiation

Although the current consensus of the field gravitates toward the compensation view when interpreting the functional role of the PASA model [9], the alternative view is that the age-related increase in brain activation is due to a loss of functional specificities. This concept is called de-differentiation, which is conceptually the reversal process of functional differentiation that occurs during the developmental period [40]. According to this view, the PASA model is a representation of functional degrading that does not necessarily have a compensatory effect. The results from the current study cannot provide direct evidence to support either view because the group difference between high- and low-performing older adults did not show significant differences. The increase of the prefrontal-hippocampus functional connectivity found in the low-performing older adults may be interpreted as a trace of the compensation because usually it is associated with better memory performance [31], but from the behavioral results it is no longer helping. To summarize, the current study did not yield convincing evidence that supports either the compensation view or the differentiation view.

### 4.4. Cerebellar Participation

We observed cerebellum activation in both novelty and relational processing. This is consistent with past studies showing cerebellar involvement in episodic encoding [41,42,43,44]. We found right inferior cerebellar activity only in older adults and not young adults for novelty processing. In contrast, greater bilateral superior and right inferior cerebellum activation was observed in young adults for relational processing. It has been suggested that the right superior cerebellum is associated with encoding success for semantic information of relational processing and the bilateral superior cerebellum with non-semantic information of letters [43]. Specifically, right Crus 1 was found to be associated with the retrieval of episodic memory [41]. Our findings provided further evidence for the involvement of the cerebellum in episodic memory.

### 4.5. Other Thoughts on the Analysis Method

We demonstrated in this study that task performance during fMRI can adequately separate the older adults into low and high performers. Conventionally, this separation would require separate standardized cognitive tests. Our approach ensures that we examine the actual difference directly related to the task without relying on the assumption that individuals who scored better on cognitive tests should perform better during the fMRI task. The current approach offers an opportunity for the experimenters of past studies to further examine their findings based on participants’ fMRI task performance.

### 4.6. Limitations and Future Directions

The older adults in the CBL dataset showed more difficulty in recognizing non-scrambled novel scenes during the post-scan recognition test. This suggests they were less engaged during the novelty-related processing when recognizing indoor/outdoor scenes inside the MRI scanner. The lack of the expected results may be related to poor engagement during the task. When we interpret the results from the current study, this limitation should be kept in mind.

We did not control for the number of low- and high-performing older adults when separating them into the two subgroups. As a result, we had more high performers than low performers, which may have reduced the statistical power to detect the group difference. Furthermore, this subgrouping is inherently affected by age [45]. In this study, low-performing older adults were older than the high-performing older adults. However, no significant difference in the MMSE score was found between low and high-performing older adults, justifying that both performance groups are within normal cognitive limits. Hence neither age nor MMSE score were submitted as covariates in our analyses.

We did not delineate the hippocampus into its anterior and posterior regions. Studies by Ta et al. [11,14] have shown functional specialization within the hippocampus, with the anterior hippocampus responsible for the relational processing of scenes and the posterior hippocampus as the neural correlate for the novelty processing of scenes. Age-related vulnerability to structural atrophy and activation was also demonstrated, with the anterior hippocampus found to be more prone to age-related changes [14]. It remains unclear how the anterior-posterior hippocampal difference may interact with the current results.

Finally, for future studies, it would be worthwhile to examine if the PASA model and compensatory view of age-related changes in functional activation and connectivity are supported for encoding other types of stimuli, such as words, faces, or abstract patterns.

## 5. Conclusions

The present study provided partial support for the PASA model for scene encoding. In the relational processing, age-related changes in activations and functional connectivity between IFG (i.e., top-down modulation) and hippocampal/parahippocampal structures (i.e., bottom-up processing) were identified to be performance-related for the old adults. However, the expected activation increase in IFG in older adults was not observed. Thus, the posterior aspect of the PASA model was supported, but the anterior aspect of it was not directly supported by the BOLD signal increase but instead indicated by the increase of parahippocampal-IFG functional connectivity. The current study did not provide direct support for the compensatory view of the PASA model.

## Figures and Tables

**Figure 1 sensors-23-04107-f001:**
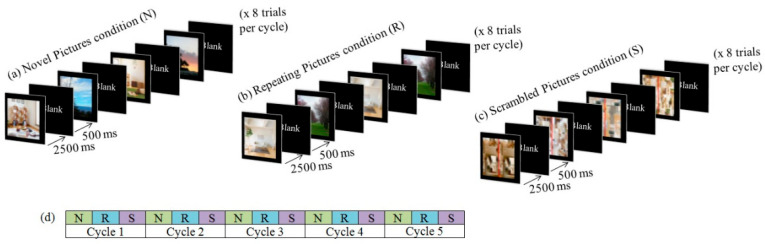
Stimulus presentation to each participant was blocked by the condition in the order of (**a**) Novel Pictures condition, (**b**) Repeating Pictures condition, and (**c**) Scrambled Pictures condition for 5 cycles. (**d**) Each block lasted 24 s and one run took 360 s in total.

**Figure 2 sensors-23-04107-f002:**
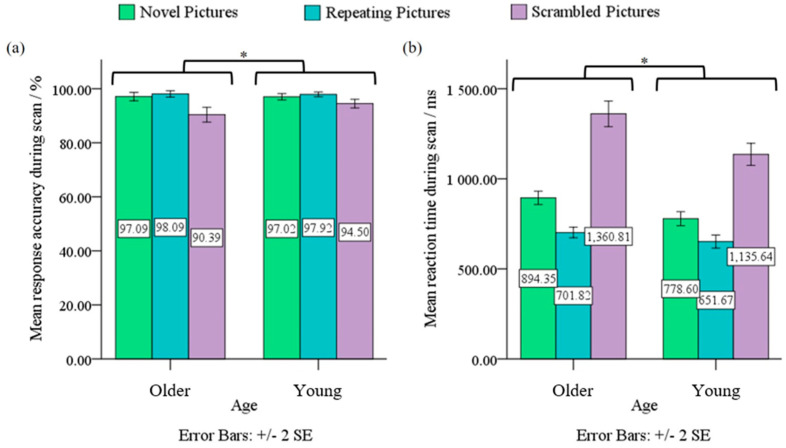
(**a**) Age × condition mixed design ANOVA on ACC revealed a significant interaction effect and a main effect of condition, at *p* < 0.05. (**b**) Age × condition mixed design ANOVA on RT revealed significant interaction effect, main effects of age and condition, at *p* < 0.05 *.

**Figure 3 sensors-23-04107-f003:**
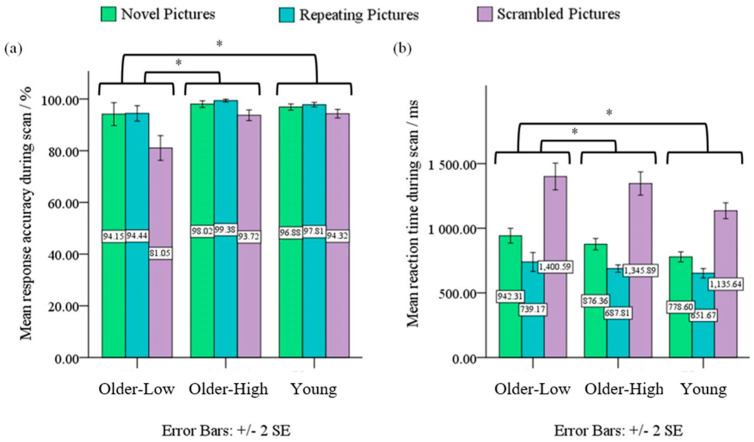
(**a**) Performance × condition mixed design ANOVA on ACC revealed significant interaction effect, main effects of performance and condition, at *p* < 0.05. (**b**) Performance × condition mixed design ANOVA on RT revealed significant interaction effect, main effects of performance and condition, at *p* < 0.05 *.

**Figure 4 sensors-23-04107-f004:**
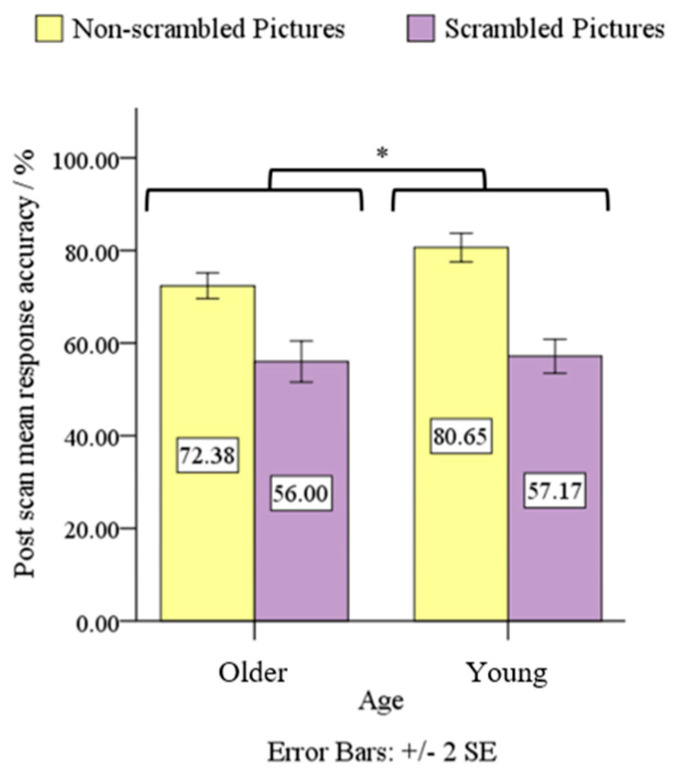
Age × condition mixed design ANOVA on post-scan ACC revealed significant interaction effect, main effects of age and condition, at *p* < 0.05 *.

**Figure 5 sensors-23-04107-f005:**
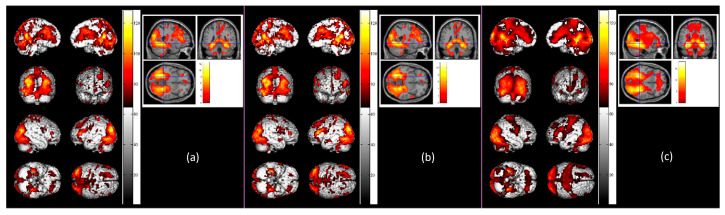
Significant cluster-level activations for novelty processing of scenes (N > R contrast), at *p* < 0.05 (*FDR*-corrected) with *k* ≥ 20, after adjusting for gray matter probability, signal-to-fluctuation-noise (SFNR), and accuracy (ACC) performance. In the figure, from the left panel: (**a**) Significant activation among older adults (*n* = 33) for novelty processing of scenes (N > R contrast); (**b**) Significant activation among older high performers (*n* = 24) for novelty processing of scenes (N > R contrast); (**c**) Significant activation among young adults (*n* = 48) for novelty processing of scenes (N > R contrast).

**Figure 6 sensors-23-04107-f006:**
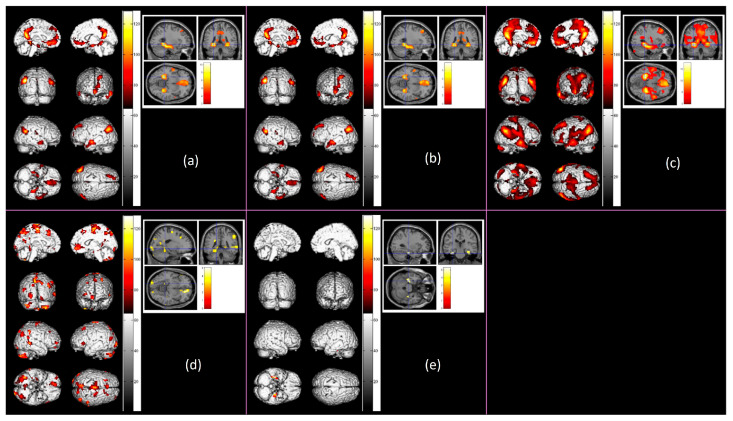
Significant cluster-level activations for relational processing of scenes (N > S contrast), at *p* < 0.05 (*FDR*-corrected) with *k* ≥ 20, after adjusting for gray matter probability, signal-to-fluctuation-noise (SFNR), and accuracy (ACC) performance. In the figure, from the top left panel: (**a**) Significant activation among older adults (*n* = 33) for relational processing of scenes (N > S contrast); (**b**) Significant activation among older high performers (*n* = 24) for relational processing of scenes (N > S contrast); (**c**) Significant activation among young adults (*n* = 48) for relational processing of scenes (N > S contrast); (**d**) Significant activation for young versus older adults (*n* = 81) for relational processing of scenes (N > S contrast); (**e**) Significant activation for young versus older low performers (*n* = 57) for relational processing of scenes (N > S contrast).

**Figure 7 sensors-23-04107-f007:**
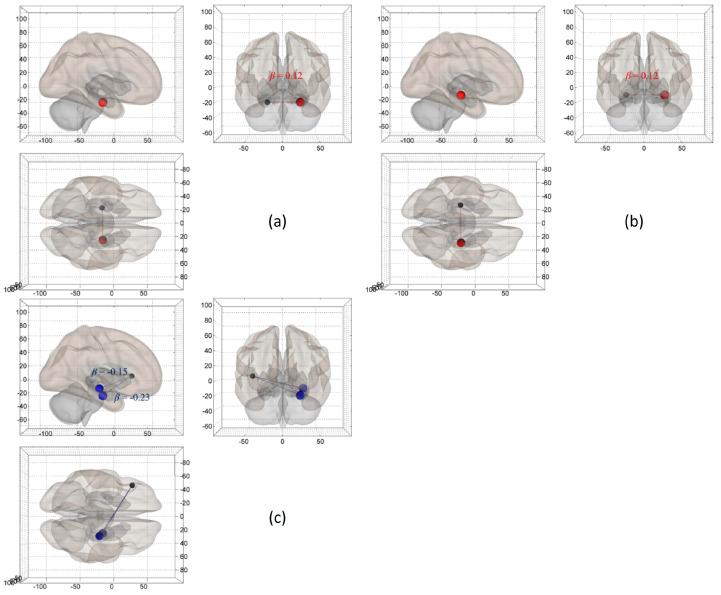
Significant ROI-to-ROI functional connectivity for relational processing of scenes (N > S contrast), at *p* < 0.05 (*FDR*-corrected), after adjusting for intracranial brain volume and ACC performance. N: Novel; S: Scrambled. The in-plot descriptions, top left, (**a**) Significant positive left parahippocampus-right parahippocampus functional connectivity among older low performers (*n* = 6) for relational processing of scenes (N > S contrast); Top right, (**b**) Significant positive left hippocampus-right hippocampus functional connectivity among young performers (*n* = 25) for relational processing of scenes (N > S contrast); Bottom, (**c**) Significant negative left IFG-right hippocampus/parahippocampus functional connectivity for young versus older low performers (*n* = 31) for relational processing of scenes (N > S contrast).

**Table 1 sensors-23-04107-t001:** Significant cluster-level activations for novelty encoding of scenes (N > R contrast), at *p* < 0.05 (*FDR*-corrected) with *k* ≥ 20.

Brain Region		Hemisphere	BA	Cluster Size	Peak Activation		MNI Coordinates		
					*p*-Value (*FDR*-Corrected)	*t*-Value	x	y	z
Old adults									
	Superior frontal gyrus	R	6	37	<0.01	3.49	21	−6	75
	Precentral gyrus	R	6	22	<0.01	3.70	63	3	26.25
					<0.05	2.38	63	6	15
			4	34	<0.05	3.16	45	−9	60
					<0.05	2.68	39	−21	67.5
	Postcentral gyrus	R	1		<0.05	2.41	54	−18	56.25
	Superior temporal gyrus	L	21	45	<0.01	3.25	−54	−9	−15
			22		<0.01	3.24	−63	3	−11.25
	Middle temporal gyrus	L	21		<0.05	2.86	−60	6	−22.5
	Fusiform gyrus	L	19	13,263	<0.001	12.51	−27	−60	−7.5
	Parahippocampal gyrus	L	36		<0.001	11.74	−24	−42	−7.5
		R			<0.001	10.54	33	−33	−15
	Cerebellum (Cbl) Lobule 9	R		106	<0.001	4.83	21	−42	−48.75
Old-High adults									
	Middle frontal gyrus	R	6	23	<0.05	3.28	30	18	63.75
					<0.05	2.80	33	27	56.25
	Precentral gyrus	R	6	104	<0.01	3.78	60	3	26.25
			4		<0.01	3.60	69	−9	30
			6		<0.01	3.57	66	9	15
			4	40	<0.05	3.05	45	−12	60
	Postcentral gyrus	R	3		<0.05	3.30	36	−24	71.25
					<0.05	2.56	48	−24	63.75
	Superior temporal gyrus	R	22	46	<0.005	4.02	63	0	−11.25
					<0.005	3.97	63	9	−11.25
	Middle temporal gyrus	L	21	77	<0.01	3.72	−63	0	−15
					<0.01	3.64	−54	−15	−18.75
					<0.05	3.42	−60	6	−22.5
	Fusiform gyrus	L	19	12,670	<0.001	11.61	−27	−63	−7.5
	Hippocampus	R			<0.001	10.12	33	−30	−15
	Parahippocampal gyrus	L	36		<0.001	10.03	−24	−42	−7.5
	Cbl Lobule 9	R		48	<0.005	3.94	21	−42	−48.75
	Cbl Lobule 8				<0.05	2.78	33	−48	−52.5
	Cbl Louble 9				<0.05	2.61	15	−54	−45
Young adults									
	Middle frontal gyrus	R	46	38	<0.005	3.32	51	42	15
	Fusiform gyrus	R	37	23,638	<0.001	16.38	33	−45	−15
	Hippocampus	R			<0.001	15.79	36	−36	−11.25
	Parahippocampal gyrus	L	36		<0.001	18.01	−30	−42	−7.5

**Table 2 sensors-23-04107-t002:** Significant cluster-level activations for relational encoding of scenes (N > S contrast), at *p* < 0.05 (*FDR*-corrected) with *k* ≥ 20.

Brain Region		Hemisphere	BA	Cluster Size	Peak Activation		MNI Coordinates		
					*p*-Value (*FDR*-Corrected)	*t*-Value	x	y	z
Old adults									
	Superior frontal gyrus	L	9	563	<0.001	5.66	−12	60	11.25
	Superior temporal gyrus	R	39	199	<0.001	7.10	57	−57	22.5
		L	38	226	<0.001	7.10	−54	−3	−11.25
					<0.005	4.71	−45	9	−30
		R	22	86	<0.005	4.26	60	3	−11.25
	Middle temporal gyrus	L	39	302	<0.001	9.06	−42	−72	33.75
			21	226	<0.01	3.97	−54	−18	−18.75
	Angular gyrus	L	39	302	<0.001	9.53	−42	−75	41.25
	Middle temporal gyrus	R	21	86	<0.005	4.40	51	−9	−18.75
	Insula	R	13	95	<0.001	5.24	39	−18	18.75
		L		27	<0.005	4.22	−36	−24	22.5
	Cingulate gyrus	L	31	716	<0.001	5.57	3	−60	30
	Anterior cingulate	L	24	563	<0.001	7.64	0	24	−15
			32		<0.001	6.05	0	51	−11.25
	Posterior cingulate	L	30	716	<0.001	9.85	−9	−54	15
		R	23		<0.001	8.00	12	−54	18.75
	Hippocampus	L		372	<0.001	8.86	−30	−27	−15
	Parahippocampal gyrus	L	36	372	<0.001	10.45	−27	−42	−7.5
		R		325	<0.001	7.74	30	−45	−7.5
			35		<0.001	7.14	30	−30	−15
			28		<0.001	6.25	21	−12	−18.75
Old-High adults									
	Superior frontal gyrus	L	9	590	<0.001	6.91	−12	60	11.25
	Superior temporal gyrus	R	39	183	<0.001	8.43	57	−57	22.5
		L	38	210	<0.001	6.68	−54	−3	−11.25
					<0.005	4.67	−45	9	−30
		R	38	93	<0.005	4.72	60	0	−15
	Middle temporal gyrus	L	39	296	<0.001	8.76	−45	−72	33.75
			21	210	<0.005	5.19	−54	−12	−18.75
	Inferior temporal gyrus	R	20	93	<0.01	4.22	51	−9	−22.5
	Angular gyrus	L	39	296	<0.001	8.98	−42	−75	41.25
	Superior occipital gyrus	L	19	296	<0.001	9.73	−42	−81	33.75
	Insula	R	13	76	<0.005	5.48	39	−18	18.75
		L	13	27	<0.01	4.34	−36	−18	18.75
					<0.05	4.06	−42	−12	15
	Cingulate gyrus	L	31	642	<0.005	5.62	3	−60	30
	Anterior cingulate	L	32	590	<0.001	7.12	−9	48	−3.75
					<0.001	6.90	−3	51	−11.25
		R	24	25	<0.01	4.36	3	27	11.25
	Posterior cingulate	L	23	642	<0.001	9.37	−9	−57	18.75
		R	23	642	<0.001	7.77	12	−54	18.75
	Parahippocampal gyrus	L	36	337	<0.001	8.75	−27	−42	−7.5
					<0.001	7.66	−30	−30	−18.75
					<0.001	6.44	−24	−12	−22.5
		R	36	294	<0.001	6.92	27	−45	−7.5
			35		<0.001	6.11	30	−30	−15
			28		<0.005	5.35	21	−12	−18.75
Young adults									
	Inferior frontal gyrus	R	47	30	<0.001	5.05	33	33	−11.25
	Middle temporal gyrus	L	39	13,094	<0.001	16.56	−45	−72	30
	Posterior cingulate	L	30	13,094	<0.001	16.82	−12	−54	15
	Parahippocampal gyrus	L	36	13,094	<0.001	17.72	−30	−42	−11.25
	Cbl Crus I	R		237	<0.001	7.70	45	−75	−37.5
	Cbl Crus II	L		54	<0.001	4.50	−12	−90	−37.5
		R		237	<0.001	7.58	18	−90	−33.75
	Cbl Crus I				<0.001	6.76	33	−90	−33.75
	Cbl Crus II	L		54	<0.001	4.74	−27	−90	−33.75
	Cbl Lobule 9	R		37	<0.005	3.77	6	−51	−45
					<0.005	3.63	3	−57	−52.5
Young > Old adults									
	Superior frontal gyrus	L	6	125	<0.05	3.90	−24	21	56.25
		R	6	38	<0.05	3.47	12	18	56.25
					<0.05	3.30	21	21	48.75
			10	29	<0.05	3.47	33	57	3.75
	Middle frontal gyrus	L	6	125	<0.05	3.56	−27	12	45
		R	8	38	<0.05	3.54	27	30	41.25
	Inferior frontal gyrus	L	45	33	<0.05	4.64	−60	21	0
	Medial frontal gyrus	L	8	125	<0.05	4.97	−9	27	45
		R	6	360	<0.05	4.73	6	−18	63.75
					<0.05	4.06	6	−12	52.5
	Precentral gyrus	L	6	360	<0.05	4.30	−15	−12	63.75
	Inferior parietal lobule	R	40	50	<0.05	4.21	57	−45	45
	Precuneus	L	7	229	<0.05	4.42	−9	−60	45
					<0.05	4.05	−6	−72	48.75
		R	39	58	<0.05	3.74	45	−69	41.25
	Superior temporal gyrus	R	13	22	<0.05	3.70	51	−48	18.75
	Middle temporal gyrus	L	39	30	<0.05	4.26	−48	−78	15
		R	39	58	<0.05	3.96	48	−75	33.75
			21	60	<0.05	3.92	69	−42	0
			22		<0.05	3.83	57	−45	0
			21		<0.05	3.12	69	−48	11.25
	Cuneus	R	19	229	<0.05	4.36	21	−81	41.25
		L	18	22	<0.05	3.54	−12	−84	33.75
	Lingual gyrus	L	18	58	<0.05	4.75	−27	−99	0
		R	18	25	<0.05	3.23	18	−90	−11.25
	Cingulate gyrus	L	31	22	<0.05	4.20	−24	−42	26.25
	Anterior cingulate	R	32	143	<0.05	5.05	12	51	−11.25
					<0.05	4.93	18	39	0
					<0.05	4.83	15	42	−7.5
	Posterior cingulate	L	30	35	<0.05	3.96	−24	−57	15
					<0.05	3.54	−6	−66	18.75
	Parahippocampal gyrus	R	36	31	<0.05	4.13	30	−33	−15
			30	33	<0.05	4.06	24	−45	11.25
					<0.05	3.22	18	−42	0
		L	30	48	<0.05	3.66	−18	−42	−7.5
					<0.05	3.55	−15	−42	0
		R		21	<0.05	3.52	33	−15	−22.5
	Fusiform gyrus	L		48	<0.05	3.78	−21	−42	−15
	Cbl Crus I	R		25	<0.05	3.34	21	−87	−18.75
	Cbl Lobule 7b	R		180	<0.05	3.95	30	−75	−48.75
	Cbl Crus II	L		86	<0.05	3.67	−30	−75	−45
					<0.05	3.61	−39	−72	−37.5
					<0.05	3.57	−27	−87	−37.5
	Cbl Lobule 8	R		180	<0.05	4.53	36	−54	−52.5
	Cbl Crus I				<0.05	4.33	39	−63	−41.25
	Cbl Lobule 9			33	<0.05	4.00	6	−60	−52.5
Young > Old-Low adults									
	Parahippocampal gyrus	L	36	25	<0.05	5.32	−30	−24	−26.25
		R	35	49	<0.05	4.62	30	−30	−18.75
			36		<0.05	4.42	36	−27	−26.25
					<0.05	4.34	36	−24	−18.75

**Table 3 sensors-23-04107-t003:** Significant ROI-to-ROI functional connectivity for relational encoding of scenes (N > S contrast), at *p* < 0.05 (*FDR*-corrected).

ROI-ROI		*β*	*p*-Value (*FDR*-Corrected)	*t*-Value
Old-Low adults				
	Left parahippocampus–Right parahippocampus	0.12	<0.05	5.83
Young adults				
	Left hippocampus–Right hippocampus	0.12	<0.05	2.92
Young > Old-Low adults				
	Left IFG–Right hippocampus	−0.15	0.06	−2.05
	Left IFG–Right parahippocampus	−0.23	0.06	−2.20

## Data Availability

Data is available upon request from the authors.

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
