# Peer review of "Age-Related Changes in Episodic Processing of Scenes: A Functional Activation and Connectivity Study"

_sensors, 2023, doi:10.3390/s23084107_

Round 1

Reviewer 1 Report

The article is well-written and organized, but I would like to advise you about some points that would enhance your manuscript:

1) Abbreviations are not consistently defined:

a) In the abstract, IFG is not explained in the first occurrence (line 23) but in the second (line 26)!

b) PASA abbreviation is defined several times, first in line 88 and then (in the same paragraph!) in line 93, and then also in lines 155 and 779

c) MMSE is not defined at all

d) ACC is defined twice (lines 200 and 278)

e) BA - an abbreviation that labels the third column in Tables 1 and 2 is not defined

2) Figs. 5 and 6 are divided into several panels (a), (b) ... . Short description of each panel is displayed in the image with white letters on a black background, which is difficult to read, especially in the printed version. I suggest displaying these descriptions with black letters on a white background, like in Fig. 7.

3) Why do you denote parameter eta squared in ANOVA  results as eta2 and not eta2?

4) Some long sentences with a lot of subsentences are sometimes difficult to comprehend.  I will mention here all three cases where results in Tables are introduced in your manuscript:

a) Table 1, lines 457-9:  
"Supra-threshold cluster-level activations, at p < 0.05 (FDR-corrected) with k ≥ 20, for N > R contrast representing novelty processing of scenes, is shown in Table 1."

You started with the plural "activations" and end with the singular "is shown in Table 1."  I would re-write this sentence :

Results of supra-threshold cluster-level activations, at p < 0.05 (FDR-corrected) with k ≥ 20 for N > R contrast representing novelty processing of scenes, are shown in Table 1.

b) Table 2, lines 481-2, I suggest a similar correction as above:

Results of supra-threshold cluster-level activations, at p < 0.05 (FDR-corrected) with k ≥ 20 for N > S contrast representing relational processing of scenes, are shown in Table 2.

c) Table 3, lines, 522-5:
"ROI-to-ROI functional connectivity analyses were conducted with a priori ROIs, namely left and right IFG, hippocampus and parahippocampus, at p < 0.05 (FDR-corrected), for N > S contrast representing relational processing of scenes, is shown in Table 3."

I suggest replacing it with:

ROI-to-ROI functional connectivity analyses were conducted with a priori ROIs, namely left and right IFG, hippocampus, and parahippocampus at p < 0.05 (FDR-corrected) for N > S contrast representing relational processing of scenes. Results are shown in Table 3.

Reviewer 2 Report

In this paper, the authors present functional activation and connectivity analyses that were applied to examine the age-related changes on the inferior frontal gyrus, hippocampus, and parahippocampus among low/high-performing older adults and young adults. Significant parahippocampal activation was generally found in both older (high-performing) and young adults for novelty and relational processing of scenes. Younger adults had significantly greater IFG and parahippocampal activation than older adults and greater parahippocampal activation compared to low-performing older adults for relational processing – providing partial support for the PASA model. This article is clear, concise, and suitable for the scope of the journal. Several small suggestions are supplied:
1. Suggest the authors supply more detail about Age x condition mixed design ANOVA on ACC revealed significant interaction effect and the main effect of condition.
2. Suggest the authors supply more detail about significant ROI-to-ROI functional connectivity for relational processing of scenes.
